# Influence of Palliative Care Qualifications on the Job Stress Factors of General Practitioners in Palliative Care: A Survey Study

**DOI:** 10.3390/ijerph192114541

**Published:** 2022-11-05

**Authors:** Sophie Peter, Anna Maria Volkert, Lukas Radbruch, Roman Rolke, Raymond Voltz, Holger Pfaff, Nadine Scholten

**Affiliations:** 1Faculty of Human Sciences, University of Cologne, 50933 Cologne, Germany; 2Faculty of Medicine, University of Cologne, 50933 Cologne, Germany; 3Institute of Medical Sociology, Health Services Research and Rehabilitation Science, University Hospital Cologne, 50933 Cologne, Germany; 4Department of Palliative Medicine, University Hospital Bonn, 53127 Bonn, Germany; 5Department of Palliative Medicine, Medical Faculty, RWTH Aachen University, 52074 Aachen, Germany; 6Department of Palliative Medicine, University Hospital Cologne, 50933 Cologne, Germany; 7CIO Aachen Bonn Cologne Düsseldorf, 50937 Cologne, Germany

**Keywords:** general practitioner, palliative care, qualification, job stress

## Abstract

Due to demographic change, the number of patients in palliative care (PC) is increasing. General Practitioners (GPs) are important PC providers who often have known their patients for a long time. PC can be demanding for GPs. However, there are few studies on the job stress factors of GPs performing PC and the potential influence of their PC training. To get more insights, a postal survey was performed with GPs in North Rhine, Germany. The questionnaire was based on a literature search, qualitative pre-studies, and the Hospital Consultants’ Job Stress & Satisfaction Questionnaire (HCJSSQ). Participants state that a high level of responsibility, conflicting demands, and bureaucracy are the most important stressors they experienced in PC. The influence of PC qualification level on their perceived job stress factors is low. Only advanced but not specialist qualification shows a correlation with renumeration-related stress. Gender and work experience are more dominant influences. In our study, female GPs and physicians with more work experience tend to be more stressed. In conclusion, organisational barriers, such as administration, should be reduced and renumeration should be increased to facilitate the daily work of GPs.

## 1. Introduction 

Increasing life expectancy combined with an aging population and declining birth rates (demographic change) poses challenges to healthcare systems worldwide. A rapid increase in the number of people in need of care is anticipated. Outpatient healthcare in particular will be in greater demand as a result [1,2]. As a logical consequence, there is a growing number of patients with palliative care (PC) needs due to demographic change [3]. PC is defined by the International Association for Hospice and Palliative Care as “the active holistic care of individuals across all ages with serious health-related suffering due to severe illness and especially of those near the end of life. It aims to improve the quality of life of patients, their families and their caregivers” [4]. PC is provided in outpatient as well as in inpatient settings [4]. PC providers, such as nurses, physicians, and other health care providers, receive basic or specialist training [4]. While studies report that 69–82% of dying patients are in need of PC [5] and 40.7–96.1% of dying patients might benefit from PC [6], only 14% of the people in need receive PC [3]. Early delivery of PC can reduce the overall use of health care services and avoid unnecessary hospital admissions for PC patients [3].

PC providers experience many burdensome challenges in their work with dying patients [7], which can lead to job stress or high workload. Both can negatively affect patients’ treatment [8,9,10]. Potential stressors that are reported are organisational factors (e.g., poor cooperation, lack of time), patient-related factors (e.g., emotional stress because of patients’ and relatives’ demanding needs), and inefficient self-care [8,11,12,13]. Personal factors such as age and gender may also affect perceptions of job stress [14,15,16].

German PC is in an advanced stage of integration (category 4b) in international comparison, which is the highest ranking according to the Global Atlas of Palliative Care [17]. Since they often know their patients well or for a long time and have a providing and coordinating role, general practitioners (GPs) are central basic PC providers at the primary care level [7,18]. While GPs see PC as part of their job, they tend to have insufficient knowledge and training in PC [18,19,20]. In Germany, GPs can attend curricular PC trainings on a voluntary basis. The trainings qualify GPs to provide outpatient PC services. In Germany, GPs bill their PC services to the statutory health insurance funds. By completing the training courses, the GPs have the opportunity to bill more PC services with specific billing codes [21]. There are different levels of German outpatient PC qualification. The lowest level of training corresponds to GPs without additional PC training after medical school. A higher level is a basic PC qualification including a 40-h PC training course. This qualification is followed by accreditation as a physician for “specially qualified coordinated palliative medical care” (BQKPMV, German: Besonders Qualifizierte und Koordinierte Palliativmedizinischen Versorgung). Physicians have to prove they treated at least 15 PC patients or did an internship in a PC facility for at least two weeks. They also have to attend a specified training course. The highest qualification is being a PC specialist or qualified PC physician (German: qualifizierter Palliativarzt). In addition to the required training courses (extent: 120 h), the physicians have to pass an examination of the state medical association [21,22]. The PC qualification modules address how to deal with some of the PC-related stress factors. Contents taught include, for example, being involved with the physical or emotional suffering of patients, having to break bad news to patients and their relatives, symptom control, and managing death and dying well for patients. The higher the qualification level, the more detailed the topics that can be taught [22].

The idea behind the different qualifications is that physicians can treat more complex and difficult palliative cases the better qualified they are. This leads to the question of whether palliative qualification determines the perceived job stress factors among GPs. PC-related job stress factors of GPs are not well researched. Most studies about job stress factors of PC providers focus on inpatient settings and mostly but not exclusively on specialised PC facilities and wards [11,23,24,25]. We hypothesise that GPs with lower PC qualifications perceive PC-related job stress factors as more burdensome because they are less well trained [26,27,28].

## 2. Materials and Methods

Methods and results are reported based on the STROBE Statement [29] (s. Appendix A: STROBE Statement—Checklist of items that should be included in reports of cross-sectional studies). This data collection was part of the study “Evaluation of (Specialised) palliative home care in North Rhine” funded by the Innovation Funds of the statutory health insurance (acronym: APVEL, grant no.: 01VSF16007; German Clinical Trials Registration: DRKS00014748,). The general aim of the APVEL study was to evaluate palliative home care in North Rhine, Germany. The researchers conducted a postal survey on GPs’ experiences in PC in July/August 2018 (*n* = 2154 GPs; response rate: 20.6%). The questionnaire was developed based on results of qualitative data analysis (expert interviews, focus group) and a literature search. Nine items of the questionnaire are part of an already existing scale, the Hospital Consultants´ Job Stress & Satisfaction Questionnaire (HCJSSQ) [30], which was adapted for the outpatient setting and translated into German by the research team. The HCJSSQ’s reliability and validity were tested by the Cancer Research UK London Psychosocial Group [30]. Five items (“high level of responsibility”, “conflicting demands”, “uncertainty about renumeration”, “recourse” [if an audit by the Association of Statutory Health Insurance Physicians shows that a physician has prescribed uneconomically, funds can be reclaimed], and “bureaucratic burden”) were added by the research team based on the qualitative pre-studies. GPs could state on a four-point scale the extent to which they have been stressed in the past few months by the factors (1: “not at all”, 2: “a little”, 3: “quite a bit”, 4: “a lot”). 

The ethical review committee of the Medical Faculty of the University of Cologne approved the study (no. 17-297). GPs working in a medical practice in North Rhine were recruited for the study. Six GPs pre-tested the questionnaire according to content and wording. The survey was sent by mail to GPs in North Rhine, Germany, as an example region. Participants received a cover letter with study information. The cover letter described the aim of the survey and indicated that the data were collected anonymously. It also included information about the study’s funding and the research institute collecting the data. GPs were told that they provided an implicit consent to the study’s modalities by returning their questionnaire in an enclosed, self-addressed, and stamped envelope. Data collection included three reminders according to Dillman’s Total Design Method [31]. 

Statistical analysis was performed using descriptive and multivariate analysis in Stata 15. We performed the Shapiro-Wilk tests to check the assumption of normality of the distribution of the means. Beside the variables “Dealing with patients or relatives having expectations of care that cannot be met” and “Having to break bad news to patients and their relatives”, all means of the dependent variables have normal distributed means. This is the reason why we carried out linear regression analyses to examine job stress factors experienced by GPs related to PC [32,33,34]. Based on the literature, we included the following as independent variables in the 14 models: qualification, work experience, the assumption that PC is GPs’ duty, having sufficient knowledge about PC, the assumption that GPs can provide adequate PC, age, and gender. To avoid bias, we excluded GPs who additionally work on specialised PC teams from the analyses. Cases with missing values in the previously named items were also excluded from the analyses.

## 3. Results

Sociodemographic factors are shown in Table 1 (s. Table 1).

Having a high level of responsibility, conflicting demands on working time (e.g., patient care, administration, research), bureaucratic burden, and being involved with the emotional distress were the most stressful factors in GPs’ PC according to the participants. Further ranking is shown in Table 2 (s. Table 2). 

The regression analyses showed only one significant correlation between these job stress factors and palliative qualifications of the GPs (s. Table 3: Linear Regression Models for different job stress factors in PC, shortened version; detailed version of the tables. Appendix A: Linear Regression Models for different job stress factors in PC, detailed version). Having a BQKPMV qualification is, in comparison with GPs without a PC qualification, significantly correlated with uncertainty about remuneration (recourse). Feeling stressed because of being involved with the physical suffering of patients is positively associated with work experience and female gender. Also, perceived stress due to being involved with the emotional distress of patients is positively correlated to work experience and female gender. Job stress due to dealing with patients or relatives having expectations of care that cannot be met is negatively correlated with the assumption that GPs can provide adequate PC. Finally, perceived job stress based on dealing with angry or blaming patients/relatives, having to break bad news to patients and their relatives, managing death and dying well for patients, and bureaucratic burden are positively correlated with being a female GP. 

## 4. Discussion

In this study, we were able to identify job stress factors of German GPs in their PC provision. As far as we know, this is one of few studies on job stress factors of outpatient PC physicians. The survey was based on the HCJSSQ [30] and self-developed items. Despite our hypothesis, PC qualification only partly influenced the experienced job stress factors in PC. The most frequent and significant correlations of PC-related job stress factors were found with work experience in years and gender. 

### 4.1. Most Important Stressors

GPs experience having a high level of responsibility, conflicting demands on their working time (e.g., patient care, administration, research), and bureaucratic burden as the most stressful aspects in PC. These three factors stress GPs “quite a bit.” Our findings tend to indicate that organisational factors are rated as more stressful by the participants. A study by Ding et al. (2022) showed different results: GPs in Australia stated that the psychological and physical treatment of patients were the most stressful tasks in PC [35]. However, in a study on job stress of nurses by Peters et al. (2012), organisational factors and the working environment were also identified as important stress factors [13]. The influence of workload on the work stress of physicians in hospitals has also been shown in a previous study [36], and bureaucracy was one of the most important barriers for PC provision according to a systematic review by Carey et al. (2019) [19]. 

### 4.2. Influence of PC Qualification on Job Stress Factors

In recent studies on health care professionals’ job stress, the level of qualification is often discussed as an important factor influencing the perception of stress [37,38,39]. Most studies revealed that a higher qualification is a protective factor against job stress [26,27,28]. However, Diehl et al. (2019) reached an opposite conclusion in their study: PC-qualified nurses were more stressed than less qualified nurses [40]. Our results are similarly directed, but PC qualifications seem to have little influence on PC-associated job stress of GPs, as BQKPMV qualification significantly correlates with only one stress factor (renumeration). This could be related to increased awareness of the patients’ suffering with higher qualification [41], so that GPs with higher qualification are more competent but still feel more stressed. On the contrary, this could also indicate an inadequate level of training required for the PC qualifications, which did not really have an impact on GPs’ attitudes. 

### 4.3. Influence of Work Experience on Job Stress Factors

In our study, having more work experience is positively correlated with the feeling of being stressed because of involvement in the emotional and physical distress of patients. Parizad et al. (2021) showed similar results in their paper according to the job stress of nurses. More work experience was associated with higher job stress [42]. However, most studies on the subject show an opposite correlation, namely that less work experience of health care practitioners is associated with higher levels of job stress [43,44]. Ang et al. (2018) even describe more work experience as being associated with being able to cope better with job stress [15]. The fact that we found opposite results may be due to the research subject of PC in combination with awareness of one’s own mortality, the accompanying empathy for patients, and increasing professional experience [45]. Again, as with qualification level, feeling more stressed with more work experience could be related to increased awareness of the problems and suffering related to life-threating illnesses with more work experience [46]. 

### 4.4. Influence of Gender on Job Stress Factors

Our results show that female gender correlates with more pronounced job stress perception of GPs in PC according to bureaucracy, dealing with angry or blaming patients/relatives and breaking bad news, and managing death and dying well. Female gender is also positively correlated with feeling stressed because of being involved in the emotional and physical distress of patients. A reason may be that women show higher levels of emotional exhaustion [15]. Kim et al. (2020) also showed higher job stress for health care practitioners in inpatient settings [47]. Other studies showed no gender differences in the experiences of job stress [48,49]. Higher work stress in female GPs could also not represent a gender difference, but rather reporting bias, as women may be more likely to report job stress compared to men [16]. 

### 4.5. Strengths and Limitations

Even though the study’s data were collected in 2018, the issue of job stress in PC is still relevant. The evaluation was not confounded by pandemic-related stressors. It can be assumed that, related to lockdown or increased workload in the pandemic, PC-related stress will have increased considerably in the last two years [50]. Working in PC involved stressors for health care providers even before COVID-19 [51,52]. Recent studies still indicate that health care providers in PC experience job stress [7,53], which underlines that our results address a problem that is still topical and relevant.

In 2018, the BQKPMV was newly introduced and was therefore associated with many uncertainties in renumeration and recourse. At this point in time, this qualification level might not represent GPs having been trained for the new qualification, but rather those GPs who already provided PC getting licensed for the new qualification with the specific expectation to get adequate renumeration for what they already did previously. It is not surprising, therefore, that renumeration is seen as a job stress factor in the BQKPMV. The response rate of this study is average for German GP surveys [54]. One strength is that the questionnaire was based on the literature and qualitative pre-studies to adapt it to German PC provision and especially to the PC provision in North Rhine. While the data give only a regional overview of GPs’ job stress factors in PC, North Rhine is in the German average for many quality indicators for PC [55]. North Rhine consists of urban and rural parts and can be seen as a model German region. GPs who are additionally working on specialised PC teams were excluded from the analyses to avoid bias based on their severely ill patients’ treatment. However, this might have prevented recognition of meaningful differences, as this level of GP participation in PC might have been the only significant level of training and qualification. Basic level and the BQKPMV might not require adequate training to make a difference in work-related stress. To avoid effects due to potential conflicts of responsibility between the provision of services in traditional family medicine and palliative care, we excluded physicians from the analyses who additionally work in specialised PC teams. GPs also rated job stress factors that are not unique to PC, such as bureaucracy. In addition, the regression models show some dropouts due to missing values. The R² for the regression models are low. Other, undiscovered determinants of job stress factors are therefore not included as variables in the models. These influencing factors can be identified through further research. 

### 4.6. Practical Implications 

As practical implications, the need for offers for stress reduction [56] such as supervision, quality circles [19], stress management training [24,57], and collegial case consultation (intervision) [19] can be deduced. According to our data, especially female GPs and physicians with more professional experience can benefit from this, while access to such offers should be open to all GPs regardless of the level of qualification. Organisational barriers such as renumeration, conflicting demands, and bureaucracy should be balanced with the need for quality assurance for GPs’ PC provision [48]. One solution strategy is also that awareness should be raised about the job stress of GPs in PC [23]. These measures can be seen as a prevention of overwork or health consequences for GPs [58]. 

## 5. Conclusions

A high level of responsibility, conflicting demands on the working time, and bureaucracy are the most important stressors that the GPs in our study experienced in PC. The influence of PC qualification level on the perceived job stress factors of participating GPs seems low. Gender and work experience are more dominant influencing factors. These should be given more attention in the orientation on PC in general practice and in PC training to possibly improve it. Organisational barriers, such as in renumeration and bureaucracy, should be reduced to facilitate the daily work of GPs to avoid negative long-term effects of job stress on GPs’ physical and mental well-being [59,60]. 

## Figures and Tables

**Table 1 ijerph-19-14541-t001:** Sociodemographic factors.

		*N* (%)
Numbers of participating GPs		445 *
Gender	Male	208 (48.7%)
	Female	217 (50.8%)
	Non-binary	0 (0%)
Age in years (Mean)		53.6 (min: 33; max: 78)
Highest PC qualification	Advanced PC qualification	37 (8.7%)
	BQKPMV	10 (2.4%)
	Basic PC qualification	159 (37.4%)
	3 months’ work experience in an inpatient PC facility	21 (4.9%)
	None	198 (46.6%)
Working experience in years (Mean)		17.8 (min: 0; max: 41)

* 18 GPs were excluded from the analyses due to additionally working in a specialised PC team.

**Table 2 ijerph-19-14541-t002:** Job stress factors of GPs in PC.

Extent That a Source of Stress Has Contributed to Your Overall Job Stress in the Past Few Months	Not at All	A Little	Quite a Bit	A Lot
Being involved with the physical suffering of patients ^1^	15 (3.75%)	213 (53.25%)	154 (38.50%)	18 (4.50%)
Being involved with the emotional distress ^1^	10 (2.50%)	160 (40.00%)	196 (49.00%)	34 (8.50%)
Dealing with angry or blaming patients/relatives ^1^	44 (11.00%)	220 (55.00%)	109 (27.25%)	27 (6.75%)
Dealing with patients or relatives having expectations of care that cannot be met ^1^	36 (9.00%)	242 (60.50%)	103 (25.75%)	19 (4.75%)
Dealing with patient/relative complaints about care you have provided ^1^	143 (35.75%)	207 (51.75%)	42 (10.50%)	8 (2.00%)
Having to break bad news to patients and their relatives ^1^	29 (7.25%)	219 (54.75%)	137 (34.25%)	15 (3.75%)
Being unable to control patients’ symptoms ^1^	40 (10.00%)	233 (58.25%)	112 (28.00%)	15 (3.75%)
Caring for patients who refuse treatment ^1^	134 (33.50%)	212 (53.00%)	47 (11.75%)	7 (1.75%)
Having a high level of responsibility ^2^	18 (4.50%)	115 (28.75%)	212 (53.00%)	55 (13.75%)
Conflicting demands on my working time (e.g., patient care, administration, research) ^2^	29 (7.25%)	114 (28.50%)	201 (50.25%)	56 (14.00%)
Manage death and dying well for patients ^1^	40 (10.00%)	226 (56.50%)	120 (30.00%)	14 (3.50%)
Uncertainty about remuneration ^2^	107 (26.75%)	184 (46.00%)	80 (20.00%)	29 (7.25%)
Inadequate remuneration (recourse) ^2^	52 (13.00%)	171 (42.75%)	135 (33.75%)	42 (10.50%)
Bureaucratic burden ^2^	20 (5.00%)	139 (34.75 %)	180 (45.00%)	61 (15.25%)

*N* = 400 (45 (10.11%) of the cases were excluded due to missing data). ^1^ Items based on the HCJSSQ [30]. ^2^ Items based on the qualitative pre-studies.

**Table 3 ijerph-19-14541-t003:** Linear regression models for different job stress factors in PC, shortened version.

		Being Involved with the Physical Suffering of Patients ^1^	Being Involved with the Emotional Distress ^1^	Dealing with Angry or Blaming Patients/Relatives ^1^	Dealing with Patients or Relatives Having Expectations of Care That Cannot Be Met ^1^	Having to Break Bad News to Patients and Their Relatives ^1^	Manage Death and Dying Well for Patients ^1^	Uncertainty about Remuneration ^2^	Bureaucratic Burden ^2^
Qualification	None (Reference)								
	Advanced PC qualification	0.138 (0.286)	0.220 (0.105)	−0.053 (0.730)	−0.019 (0.888)	−0.007 (0.958)	0.134 (0.329)	0.258 (0.135)	0.094 (0.548)
	BQKPMV	0.290 (0.169)	0.231 (0.294)	0.394 (0.113)	0.223 (0.318)	0.321 (0.140)	0.032 (0.888)	0.635 * (0.023)	0.231 (0.365)
	Basic PC qualification	0.070 (0.373)	0.149 (0.073)	−0.178 (0.057)	−0.096 (0.250)	−0.044 (0.593)	0.035 (0.674)	−0.160 (0.129)	−0.111 (0.246)
	3 months’ work experience in an inpatient PC facility	−0.237 (0.172)	−0.012 (0.947)	−0.299 (0.145)	−0.239 (0.195)	−0.309 (0.085)	−0.156 (0.399)	0.141 (0.543)	0.177 (0.399)
Work experience in years		0.013 * (0.039)	0.015 * (0.020)	0.002 (0.794)	0.002 (0.788)	0.011 (0.101)	0.008 (0.223)	−0.008 (0.313)	0.002 (0.743)
PC is GPs’ duty		−0.065 (0.357)	−0.027 (0.713)	0.025 (0.761)	−0.022 (0.767)	−0.067 (0.359)	−0.068 (0.364)	−0.164 (0.082)	−0.102 (0.232)
I have sufficient knowledge about PC		−0.039 (0.493)	−0.035 (0.556)	0.066 (0.322)	0.035 (0.554)	−0.004 (0.952)	−0.012 (0.842)	0.043 (0.565)	0.120 (0.078)
GPs can provide adequate PC		0.028 (0.580)	−0.039 (0.453)	−0.053 (0.369)	−0.112 * (0.035)	−0.045 (0.380)	−0.067 (0.211)	−0.062 (0.349)	−0.102 (0.093)
Age		−0.005 (0.451)	−0.011 (0.132)	−0.003 (0.687)	−0.005 (0.506)	−0.011 (0.136)	−0.001 (0.904)	−0.002 (0.852)	0.000 (0.960)
Gender		0.145 * (0.035)	0.207 ** (0.004)	0.194 * (0.017)	−0.077 (0.293)	0.185 ** (0.009)	0.157 * (0.032)	0.020 (0.826)	0.234 ** (0.005)
Constant		2.510 *** (0.000)	2.881 *** (0.000)	2.090 *** (0.000)	2.945 *** (0.000)	2.877 *** (0.000)	2.414 *** (0.000)	2.978 *** (0.000)	2.643 *** (0.000)
R^2^		0.042	0.048	0.044	0.033	0.048	0.033	0.069	0.051

*N* = 381 (64 (14.38%) of the cases were excluded due to missing data). *p*-values in parentheses. * *p* < 0.05, ** *p* < 0.01, *** *p* < 0.001. ^1^ Items based on the HCJSSQ [30]. ^2^ Items based on the qualitative pre-studies.

## Data Availability

The data are not publicly available due to ethical and legal restrictions, as participants of this study did not agree for their data to be shared publicly.

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
