# Peer review of "Influence of Palliative Care Qualifications on the Job Stress Factors of General Practitioners in Palliative Care: A Survey Study"

_ijerph, 2022, doi:10.3390/ijerph192114541_

Round 1
Reviewer 1 Report
We acknowledge that this is a valuable research paper on stress related to palliative care for GPs. Please consider revising or otherwise addressing the following points.
1) Please specify how many samples (N) were used in Table 2.
2) In the regression analysis in Table 3, please specify how many cases were excluded due to missing values.
3) I think we can consider a few more reasons why the longer the work experience, the higher the stress. For example, is it possible that the longer the work experience, the more empathy one feels for the patient, and therefore the more one feels the patient's distress, the more stressful it is? Please present any previous studies that suggest this.
4) Similarly, if one has more work experience, one may be at higher level in the organization and will have a position of responsibility. Is it possible that such a position of responsibility made one more susceptible to stress? Please consider this.
Author Response
We acknowledge that this is a valuable research paper on stress related to palliative care for GPs. Please consider revising or otherwise addressing the following points.
>>response: Thank you very much for your valuable feedback on our paper. I will address each of your comments individually below.
1) Please specify how many samples (N) were used in Table 2.
>>response: Thank you for the advice. I added the number of samples (N) in Table 2: “N=400” (s. tracked document, Table 2, line 147)
2) In the regression analysis in Table 3, please specify how many cases were excluded due to missing values.
>>response: Thank you for you feedback according to the missings in Table 3. There were 64 cases with missing data that we excluded from the regression analyses. I have provided the numbers in Table 3: " N=381 (64 (14.38%) of the cases were excluded due to missing data)” (s. tracked document, Table 3, line 163)
3) I think we can consider a few more reasons why the longer the work experience, the higher the stress. For example, is it possible that the longer the work experience, the more empathy one feels for the patient, and therefore the more one feels the patient's distress, the more stressful it is? Please present any previous studies that suggest this.
>>response: Thank you for you feedback on this part of the discussion. We meant it in a similar way as you describe it in your comment. I have changed the wording and adapted it to your suggestions: “The fact that we found opposite results may be due to the research subject of PC in combination with awareness of one's own mortality, the accompanying empathy for patients, and increasing professional experience [45].” (s. tracked document, lines 204-208)
4) Similarly, if one has more work experience, one may be at higher level in the organization and will have a position of responsibility. Is it possible that such a position of responsibility made one more susceptible to stress? Please consider this.
>>response: Thanks for pointing this out. I agree with you that higher work experience often leads to job position with higher responsibilities. Unfortunately, I find it difficult to transfer this thought to the organization of general practitioners' offices in Germany.
There are two ways for GP to work in an outpatient practice: Either they are employed by the practice, or they own the practice themselves. It is not mandatory that you have more professional experience as a practice owner than as an employee.
But your basic idea is one reason why we excluded GPs who additionally work in a specialised PC team from our analyses. I added this information in the “strengths and limitations” section: “To avoid effects due to potential conflicts of responsibility between the provision of services in traditional family medicine and palliative care, we excluded physicians from the analyses who additionally work in specialised PC teams.” (s. tracked document, lines 248-251)

Reviewer 2 Report
Thank you for the opportunity to review your work. I have provided some points around which clarification would be helpful to enable me to fully appraise the manuscript.
The first paragraph of the introduction may require revision. The first statement relating to demographic change could be expanded and explained further. Furthermore, refining the statement "PC is provided in all health care settings" is needed. PC may have relevance across all settings and the ideal be that it is present, but this does not reflect reality.
Please add a rationale for the delay between conducting the research in 2018 and submission in 2022. What are the implications for how well the findings reflect current practice? Could COVID have influenced scores and the extent of stress since 2018?
What proportion of data were excluded due to missingness?
Please add detail about the justification for using parametric tests with this sample and the obtained data.
Please clarify what 'Divers' are in Table 1 and why they are included.
In Table 1, the formatting of numbers against "3 months' work experience..." makes it difficult to know which data is aligned with this category
I am not clear why the mean has been used in Table 2 as using the mean for central tendency with Likert scales is very difficult to interpret or make sense of. Please can you clarify why it has been used?
For the construction of the linear regression model, is there evidence to suggest that age and gender affect job stress to warrant its inclusion? Why were these included in the models rather than adjusting for certain variables?
Author Response
1. Thank you for the opportunity to review your work. I have provided some points around which clarification would be helpful to enable me to fully appraise the manuscript.
>>response: Thank you very much for reviewing our paper and improving it with your feedback. I will address each of your comments individually below.
2. The first paragraph of the introduction may require revision. The first statement relating to demographic change could be expanded and explained further. Furthermore, refining the statement "PC is provided in all health care settings" is needed. PC may have relevance across all settings and the ideal be that it is present, but this does not reflect reality.
>>response: Thank you for your suggestions for the introduction- your feedback improved this chapter a lot!
I have expanded the paragraph on demographic change: “Increasing life expectancy combined with an aging population and declining birth rates (demographic change) poses challenges to healthcare systems worldwide. A rapid increase in the number of people in need of care is anticipated. Outpatient healthcare in particular will be in greater demand as a result [2,3]. As a logical consequence there is a growing number of patients with palliative care (PC) needs due to demographic change [4].” (s. tracked document, lines 33-38)
I also edited the sentence "PC is provided in all health care settings". I apologize for the unclear wording!: “PC is provided in outpatient as well as in the inpatient settings [5]. PC providers, like nurses, physicians, and other health care providers, receive basic or specialist training[5].” (s. tracked document, lines 43-46)
To better illustrate that care for palliative patients is not always optimal I have formulated this sentence: “While studies report that 69%–82% of dying patients are in need of PC [6] and 40.7%–96.1% of dying patients might benefit from PC [7], only 14% of the people in need receive PC [4].” (s. tracked document, lines 46-48)
3. Please add a rationale for the delay between conducting the research in 2018 and submission in 2022. What are the implications for how well the findings reflect current practice? Could COVID have influenced scores and the extent of stress since 2018?
>>response: Thank you for your questions and for raising this issue. I have further expanded the paragraph in the “strengths and limitations” section and added some more studies that prove the relevance and topicality of job stress in palliative care: “Even though the study’s data were collected in 2018, the issue of job stress in PC is still relevant. The evaluation was not confounded by pandemic-related stressors. It can be assumed that related to lockdown or increased workload in the pandemic, PC-related stress will have increased considerably in the last two years [50]. Working in PC involved stressors for health care providers even before COVID-19 [51,52]. Re-cent studies still indicate that health care providers in PC experience job stress [8,53], which underlines that our results address a problem that is still topical and relevant.” (s. tracked document, lines 224-231)
4. What proportion of data were excluded due to missingness?
>>response: Thank you for you feedback. I added the proportions of missings in Table 2 and Table 3:
“N=400 (45 (10.11%) of the cases were excluded due to missing data)” (s. tracked document, Table 2, line 147)
and
“N=381 (64 (14.38%) of the cases were excluded due to missing data)” (s. tracked document, Table 3, line 163)
5. Please add detail about the justification for using parametric tests with this sample and the obtained data.
>>response: Thank you for adding this important information to our paper. We performed the Shapiro-Wilk tests to check the assumption of normality of the distribution of the means. Beside the variables “Dealing with patients or relatives having expectations of care that cannot be met” and “Having to break bad news to patients and their relatives” all means of the dependent variables have normal distributed means. That’s the reason why we decide to use linear regression. There is a long lasting debate on the level of measurement of Likert scale items and the appropriateness of applying parametric or non-parametric analyses. We support the position that there is empirical evidence showing that parametric tests on Likert scales can also lead to robust results. I added this information and references to the relevant literature in the “methods” section: “We performed the Shapiro-Wilk tests to check the assumption of normality of the distribution of the means. Beside the variables “Dealing with patients or relatives having expectations of care that cannot be met” and “Having to break bad news to patients and their relatives” all means of the dependent variables have normal distributed means. This is the reason why we carried out linear regression analyses to examine job stress factors experienced by GPs related to PC [32–34].” (s. tracked document, lines 124-129).
6. Please clarify what 'Divers' are in Table 1 and why they are included.
>>response: Thank you for your feedback. Here we made a translation error from German to English, please excuse this. We asked which gender the participants assign themselves to. The options were “female”, “male” and “non binary”. I corrected the wording in Table 1 (s. tracked document, Table 1, line 139).
7. In Table 1, the formatting of numbers against "3 months' work experience..." makes it difficult to know which data is aligned with this category
>>response: Thank you for your advice on the format of Table 1. I totally agree with you. I edited the table and hope it is clearer now (s. tracked document, Table 1, line 139)
8. I am not clear why the mean has been used in Table 2 as using the mean for central tendency with Likert scales is very difficult to interpret or make sense of. Please can you clarify why it has been used?
>>response: Thank you for pointing this out. We understand your concerns about using the mean for central tendency measure. To avoid contributing to confusion or misinterpretation, we deleted the mean as the measurement for central tendency (s. tracked document, Table 2, line 147).
9. For the construction of the linear regression model, is there evidence to suggest that age and gender affect job stress to warrant its inclusion? Why were these included in the models rather than adjusting for certain variables?
>>response: Thank you for your question and bringing this thought to our attention. In fact, there are studies that suggest that age and gender affect job stress. Therefore, we included both factors in the regression model. I added this to the introduction: “Personal factors such as age and gender may also affect perceptions of job stress [15–17].” (s. tracked document, lines 55-56).

Round 2
Reviewer 1 Report
Thank you for your appropriate revise.